# A Single Site Population Study to Investigate CYP2D6 Phenotype of Patients with Persistent Non-Malignant Pain

**DOI:** 10.3390/medicina55060220

**Published:** 2019-05-28

**Authors:** Helen Radford, Karen H. Simpson, Suzanne Rogerson, Mark I. Johnson

**Affiliations:** 1Centre for Pain Research, School of Clinical and Applied Sciences, Leeds Beckett University, Leeds LS1 3HE, UK; helen.radford2@nhs.net; 2Centre for Neurosciences, Leeds Teaching Hospitals NHS Trust, Leeds LS1 3EX, UK; dr.karen.simpson@googlemail.com (K.H.S.); suzannerogerson@nhs.net (S.R.)

**Keywords:** pain, analgesics, cytochrome P-450 *CYP2D6*, chronic pain, codeine, pharmacogenetics, pain management

## Abstract

*Background and Objectives*: Codeine requires biotransformation by the CYP2D6 enzyme, encoded by the polymorphic *CYP2D6* gene, to morphine for therapeutic efficacy. CYP2D6 phenotypes of poor, intermediate, and ultra-rapid metabolisers are at risk of codeine non-response and adverse drug reactions due to altered CYP2D6 function. The aim of this study was to determine whether genotype, inferred phenotype, and urinary and oral fluid codeine O-demethylation metabolites could predict codeine non-response following a short course of codeine. *Materials and Methods*: There were 131 Caucasians with persistent pain enrolled. Baseline assessments were recorded, prohibited medications ceased, and DNA sampling completed before commencing codeine 30 mg QDS for 5 days. Day 4 urine samples were collected 1–2 h post morning dose for codeine O-demethylation metabolites analysis. Final pain assessments were conducted on day 5. *Results*: None of the poor, intermediate, ultra-rapid metabolisers and only 24.5% of normal metabolisers responded to codeine. A simple scoring system to predict analgesic response from day 4 urinary metabolites was devised with overall prediction success of 79% (sensitivity 0.8, specificity 0.78) for morphine and 79% (sensitivity 0.76, specificity 0.83) for morphine:creatinine ratio. *Conclusions*: In conclusion, this study provides tentative evidence that day 4 urinary codeine O-demethylation metabolites could predict non-response following a short course of codeine and could be utilised in the clinical assessment of codeine response at the point of care to improve analgesic efficacy and safety in codeine therapy. We offer a scoring system to predict codeine response from urinary morphine and urinary morphine:creatinine ratio collected on the morning of day 4 of codeine 30 mg QDS, but this requires validation before it could be considered for use to assess codeine response in clinical practice.

## 1. Introduction

Prescriptions for codeine-based drugs increased by 10.5 percent between 2011 and 2012 in U.K. primary care settings, with an associated net increase in drug ingredient costs of £10.3 million [1]. Codeine requires biotransformation to active metabolites (morphine) via the enzyme CYP2D6. Genetic polymorphism of the *CYP2D6* gene influences the activity of the CYP2D6 enzyme resulting in inter-patient variability that may pose a risk of toxicity and sub-optimal analgesic response [2,3,4,5,6,7,8,9,10,11,12,13,14,15,16,17]. The phenotypic groups of CYP2D6 activity are poor metabolisers (PM, minimal or no CYP2D6 activity); intermediate metabolisers (IM, reduced CYP2D6 activity); normal metabolisers (NM, normal CYP2D6 activity); and ultra-rapid metabolisers (UM, greater than expected CYP2D6 activity) [18,19,20].

PM and IM phenotypes are at risk of sub-optimal analgesic response to codeine due to lower than expected plasma concentrations of morphine. UM phenotypes are at risk of adverse drug reactions (e.g., respiratory depression) due to higher than expected concentrations of morphine. The prevalence in Caucasian adults is ~5–10% for PMs, ~2–11% for IMs, ~77–92% for NMs and ~2–11% for UMs [21]. Prevalence of UMs in North Africans, Ethiopians, and Arabs may be as high as 40% [22,23]. There does not appear to be differences in the prevalence of *CYP2D6* polymorphisms in pain patients and the general population [24].

Combining codeine with a drug that inhibits CYP2D6 can change the phenotype of an individual (i.e., phenocopying) due to competition for enzyme activity. For example, an NM may appear to be an IM or a PM due to inhibition of CYP2D6 by the confounding drug [25,26]. Knowledge about the magnitude of CYP2D6 inhibition for certain drugs aids prescribing [27,28,29]. Prescribing guidelines for CYP2D6 phenotypes have been published with recommendations that PM, IM, and UM phenotypes should be prescribed alternatives to codeine, tramadol, oxycodone, and nortriptyline [21,30,31]. The benefits of tailoring prescribing decisions to CYP2D6 phenotype has been demonstrated [9,14,32,33,34,35,36]. However, CYP2D6 screening is not part of current clinical practice. We previously reported that 19.9% of patients referred by primary care physicians to a secondary care specialist pain management clinic were at risk of drug interactions associated with co-prescription of analgesic prodrugs reliant on CYP2D6 and CYP2D6 inhibitors [37]. A method of inferring phenotype without the need for genotyping is needed at the point of care.

Accuracy of clinical CYP2D6 phenotyping has been explored using non-analgesic CYP2D6 prodrug drugs as urinary biomarkers of CYP2D6 phenotype. These drugs are not suitable to determine CYP2D6 phenotype in pain patients. Tramadol would be an unsuitable CYP2D6 phenotyping agent due to its strong opioid classification. Kirchheiner et al., [38] found that the ratio of urinary total codeine:total morphine from a 0–6 h urine sample post 30 mg codeine dose correlated to PM, NM, and UM phenotypes in healthy volunteers, although the study did not include any participants who were IM phenotypes.

The aim of our study was to determine whether genotype, inferred phenotype and urinary and oral codeine O-demethylation metabolites could predict codeine non-response in Caucasians with persistent pain following a short course of oral codeine.

## 2. Materials and Methods

The objectives of the study were to: (I) determine the *CYP2D6* genetic profile of codeine non-response in self-reported Caucasians with persistent pain; (II) determine whether codeine non-responsiveness differs between nociceptive and neuropathic pain states; and (III) determine whether urinary and oral codeine O-demethylation metabolites predicted phenotype and codeine response.

### 2.1. Study Design

A single site, population study for a clinical trial of a medicinal product (CTIMP) was designed (http://www.legislation.gov.uk/uksi/2004/1031/pdfs/uksi_20041031_en.pdf). The primary endpoint was codeine non-response defined as participants who did not display a reduction in pain scores of ≥30% for ‘average pain during the previous 24 h’ over a course of regular codeine therapy. In addition, *CYP2D6* genotype, urinary codeine O-demethylation metabolites, clinical response to codeine, and patient reported outcomes—including brief pain inventory and global impression of health—were recorded.

The study protocol was reviewed by the research team at Seacroft Hospital, Leeds Teaching Hospitals NHS Trust, Leeds U.K. and by two independent pain researchers. Study sponsorship was granted by the Research and Development department and by the Quality Assurance team at Leeds Teaching Hospitals NHS Trust. Ethical approval was granted by the Leeds East (Type 2, CTIMP flagged) Research Ethics Committee (REC: 08/H1307/132). The study was authorised by the U.K. Medicines and Healthcare Regulatory Agency, adopted by the National Institute of Health Research Clinical Research Network portfolio (UKCRN, ID 7230), and registered on the International Standard of Randomised Controlled Trials database (ISRCTN; Trial identification number: 16874724). Amendments to study design made after the commencement of the study were approved by the Research Ethics Committee and the Medicines and Healthcare Regulatory Agency. The amendments included addition of a poster advert to aid recruitment, removal of oral fluid testing following interim analysis and change to inclusion criteria to include ‘worst pain’ in 24 h pain score.

### 2.2. Recruitment and Selection of Participants

A sample of 131 self-reported Caucasian persistent non-malignant pain patients were recruited from the Pain Clinic at Seacroft Hospital, Leeds, U.K. from October 2009 to June 2014. Potential participants were identified from the patient database held by pain service or directly from clinic by their consultant. The research nurse (S.R.) searched the database for patients with pain of moderate severity that had persisted for at least six months and who were suitable for the prescription of World Health Organisation (WHO) step 2 analgesics such as codeine. Patients with uncontrolled or escalating pain were not eligible for inclusion in the study. Each potential participant was sent a letter of invitation, a participant information pack, and a consent form. They were contacted by telephone by the research nurse (S.R.) 5–7 days later to discuss willingness and suitability to participate in the study.

### 2.3. Procedure

Potential participants were invited to attend the pain research clinic at the hospital on three separate occasions within a 15-day period. Visit one involved consent, screening for eligibility, enrolment and analgesic washout; visit two involved collection of samples for *CYP2D6* genotyping, analysis of urinary codeine O-demethylation metabolites and commencement of codeine treatment; and visit three involved cessation of codeine treatment and measurement of study endpoints.

#### 2.3.1. Study Visit One

Potential participants attended the pain clinic for approximately 1 h. The visit commenced with a briefing about the study and taking signed informed consent prior to any trial procedures being conducted.

##### Screening for Eligibility

Participants were interviewed by the pain research nurse (S.R.) or the principal investigator (H.R.) to assess eligibility for study participation. Participants completed the modified Brief Pain Inventory short form (BPI-sf) [39] to screen for pain severity and interference. The BPI-sf has two validated elements for analysis: pain severity and pain interference. Pain severity is calculated as the mean of scores (on a 0–10 numerical rating scale) for the items ‘pain at its worst in the last 24 h’, ‘pain at its least in the last 24 h’, ‘pain on the average’, and ‘pain right now’. Pain interference reflects how pain interferes with daily activities and is calculated as the mean of the items documenting how much pain interfered with ‘general activity’, ‘mood’, ‘walking ability’, ‘normal work’, ‘relations with others’, ‘sleep and enjoyment of life’.

Inclusion criteria were: self-reported Caucasian; aged between 18–80 years; pain persisting greater than 3 months as diagnosed by a pain physician and recorded on the patient database; and worst pain in the last 24 h ≥4 points on question 3 of the BPI-sf (0 = No pain, 10 = Pain as bad as you can imagine; i.e., moderate to severe pain).

Exclusion criteria were: known sensitivity to codeine, history of experiencing intolerable side effects to opioids, persistent pain adequately controlled by weak opioids, history of recreational drug or alcohol abuse within the last two years, surgery, radiotherapy, chemotherapy, or nerve blocks less than four weeks prior to the study, pregnancy or lactation, significant anxiety or depression. Participants were excluded if they were unable to complete questionnaires in English, had participated in another clinical study within the previous four weeks, had been prescribed strong opioids that could interfere with urinalysis and who were unable to cease taking their medication for the study period. Females who were less than two years post-menopausal who were not taking adequate contraceptive precautions (i.e., an oral contraceptive, an approved hormonal implant, an intrauterine device or condoms/diaphragm and spermicide) or who had not undergone hysterectomy or surgical sterilisation (bilateral tubal ligation or bilateral oophorectomy) were excluded.

Procedures were undertaken to ensure there were no underlying medical conditions that required further investigation. A 20 mL venous blood sample was collected and analysed. Participants were excluded if they had inadequate renal function (i.e., serum creatinine as ≥130 µmol/L (females); ≥150 µmol/L (males)), liver enzymes aspartate aminotransferase or alanine aminotransferase more than twice the upper limit of normal, alkaline phosphatase more than twice the upper limit of normal, bilirubin outside of normal range, haemoglobin concentration outside normal limits, white blood cell count below the lower limit of normal or above 12 × 10^9^/L, and abnormal plasma electrolytes. A urine sample was collected in a plain sterile container for dipstick analysis for pH, specific gravity, leukocytes, nitrates, protein, glucose, ketones, urobilinogen, bilirubin, and blood using Combur Test^®^ (Roche Diagnostics, Risch-Rotkreuz, Switzerland) reactive strips. Participants with abnormal results were discussed with the pain consultant (K.H.S.) to ensure that the participant was safe to continue on the study, otherwise they were excluded. Females of childbearing potential were required to have a negative urine pregnancy test.

Participants also completed a self-report version of the Leeds Assessment of Neuropathic Pain Scale (S-LANSS) [40] to indicate the presence of neuropathic components of pain. A cut-point of ≥12 points on the S-LANSS was used to define the presence of neuropathic components of pain.

##### Analgesic Washout

At visit one, participants were asked to stop taking analgesic prodrugs requiring CYP2D6 biotransformation (codeine phosphate including combination therapies, tramadol) and CYP2D6 inhibitors identified using information from Baxter [28], Flockhart [29], and the Medicines and Healthcare Regulatory Authority [41]. Participants were allowed to continue with prescribed anti-depressants, anti-convulsants or non-steroidal anti-inflammatory drugs (NSAIDs) providing the treatment was initiated at least two weeks prior to visit one and remained at a stable dose for the study duration. Participants were allowed to continue with oral contraceptives that are a weak CYP2D6 inhibitor, and this was documented for analysis purposes.

All concomitant medication and any changes therein were recorded in the case report (data collection) form. Participants were asked not to commence any new drug therapy throughout the codeine treatment period (between visit two and visit three). Participants were dispensed 64 paracetamol tablets (500 mg) for breakthrough pain and instructed to take 1 g 4–6 hourly (maximum of 4 g in 24 h) for pain relief if required throughout the study period. Participants were asked to complete the BPI-sf and to document paracetamol consumption and any associated adverse events every day in a pain diary before retiring to bed.

#### 2.3.2. Study Visit Two

Visit two took place 48 h after visit one and lasted approximately 3 h. On arrival eligibility to continue with the study was checked (i.e., results from blood and urine tests) and they completed the BPI-sf which was used as pre-treatment baseline measure of pain severity and interference.

##### CYP2D6 Genotyping

*CYP2D6* genotyping was performed to determine activity scores from which phenotype was inferred. A 2 mL saliva sample was collected for *CYP2D6* genotyping after a 30-min oral fast. A Oragene•DNA Self-Collection all-in-one system (DNA Genotek, Ottawa, ON, Canada) was used for the collection, preservation, transportation, and purification of DNA from saliva. Samples were stored for no more than 6 months in the pain clinic at room temperature and protected from light (storage stability approximately 5 years: https://www.dnagenotek.com/ROW/pdf/PD-WP-005.pdf). Samples were transported ambient in batches to KBiosciences Laboratory, Hertfordshire UK for DNA extraction and processing. *CYP2D6* allele selection and base sequencing for identification was determined from allele frequencies for a Caucasian population from the literature [18,42]. A Kompetitive Allele Specific PCR genotyping system (KASP™) was used for DNA extraction and processing (competitive allele specific polymerase chain reaction) for *CYP2D6* alleles *1, *2, *3, *4, *5, *6, *9, *10, *41 and a Hybeacon assay for *CYP2D6* duplication.

The genetic data was received electronically on encrypted Excel spreadsheets from the KBiosciences Laboratory and transcribed—including deletions, duplications, and multiple SNPs/allele identification—into a study specific worksheet designed to facilitate the process of inferring phenotype from genotype (by H.R.). Alleles were allocated a CYP2D6 activity score using a scoring method by Gaedigk et al. (1999) [43]. Phenotype was inferred from the total sum of activity scores of both alleles and duplications using classifications reported by Crews et al. [44]. Worksheets were quality checked and countersigned (by K.H.S.) to ensure correct phenotype had been inferred from genotype.

##### Collection and Analysis of Urinary Codeine O-demethylation Metabolites

A urine sample was obtained for analysis of urinary codeine O-demethylation metabolites to confirm that no codeine had been consumed during the analgesic washout phase (i.e., at baseline). In addition, participants were instructed to collect a sample of urine in sterile a universal container 1–2 h after the morning dose of codeine on the last day of treatment to store the sample in a cool location until they returned to clinic the following day for study visit three. This sample was used to analyse urinary codeine O-demethylation metabolites when the participant was equilibrated with codeine. Samples were stored in the pain clinic at −20 °C until transported in batches to laboratories at Leeds Teaching Hospitals NHS Trust. All samples were stored frozen at −10 to −35 °C until analysis. Total urine morphine and codeine were quantified using liquid chromatography tandem mass spectroscopy (LC-MS/MS). Gradient reversed phase liquid chromatography was performed using a Shimadzu Prominence HPLC system (Shimadzu, Tokyo, Japan) fitted with a Thermo Hy-Purity C8 column (Thermo Fisher Scientific, Cheshire, UK). Drugs were detected using an ABSciex API 5000 tandem mass spectrometer (ABSciex, Warrington, UK). Quantitation was achieved using the internal standard peak area ratio method. The assay was calibrated using reference materials and deuterated internal standards purchased from LGC (Teddington, UK). All urine samples were treated with β-glucuronidase to hydrolyse glucuronide compounds prior to analysis to ensure that total morphine concentration was measured. The lower limit of morphine quantification was 50 µg/L. Lower limit of morphine detection was 10 µg/L. Results below the limit of quantification were provided with the understanding the degree of uncertainty would be high (CV >20%). Samples that contained an amount of morphine greater than the top calibration standard (500 µg/L) were diluted with deionised water and re-analysed. Urine creatinine was measured by either a Beckman LX20 automated analyser (Beckmann, High Wycombe, UK) using an alkaline picrate method or a Siemens Advia automated analyser (Siemens, Surrey, UK) using an enzymatic method. Morphine:creatinine ratios were calculated (µg morphine per mmol creatinine) to correct for differences in urine concentration.

##### Codeine Treatment

Codeine Phosphate tablets (30 mg, manufactured by TEVA U.K. Ltd. (Manufacturers Authorisation: PL0289/506IR), Phoenix Healthcare Distribution) were procured by the pharmacy department of Leeds Teaching Hospitals NHS Trust and stored protected from light below 25 °C. Participants received 28 tablets of 30 mg codeine phosphate (blister pack) and were instructed to take one 30 mg tablet orally every four hours to a maximum of four tablets in 24 h. This would guarantee that participants had completed three consecutive days of full dosing to ensure the participant was equilibrated with codeine when they provided their urine sample 1–2 h after the morning dose of codeine on the last day of treatment. Eight extra tablets were provided in the blister pack in case of loss or delayed clinic visit. Participants were administered their first dose of 30 mg codeine with 200 mL of water in the clinic (dosing day 0).

##### Collection and Analysis of Oral Fluid O-demethylation Codeine Metabolites

Participants provided an oral fluid sample in clinic two hours after their first dose of codeine using an Intercept^®^ collection pad (Orasure Tech Inc, Bethlehem, PA, USA). The collection pad was placed between the lower cheek and gums and gently rubbed back and forth along the gum line until the pad was moist. Once moist the collection pad was left in the mouth for 2 min and then removed and placed into a specimen vial containing 15% methanol in 4 mL ammonium acetate. Oral fluid samples were frozen and stored at −20 °C. Samples were transported in batches to laboratories at Leeds Teaching Hospitals NHS Trust for analysis of free codeine, morphine, norcodeine, and glucuronides of morphine and codeine. Analysis of oral fluid in the first 20 participants identified only one sample as positive for codeine metabolites. Following discussions with the laboratory undertaking the analysis, it was concluded this method was unsuitable to be used to infer CYP2D6 phenotype post codeine dosing. No further analysis was conducted and oral fluid sampling was removed from the study design.

At the end of study visit two, participants were issued with a pain diary to complete each day before retiring to bed. Participants were telephoned by the research nurse (S.R.) on the day before visit three so that adverse events could be assessed and to remind participants to collect their urine sample to bring to clinic.

#### 2.3.3. Study Visit Three

During visit three, participants returned their pain diary and completed the BPI-sf and the Patient Global Impression of Change in Health. The assessors (S.R. and H.R.) completed the Clinician Global Impression of Change Scale after reviewing the participant’s pain diary and any adverse events reported by the participant during the visit. The Patient Global Impression of Change questionnaire reflects the patient’s belief about treatment efficacy by rating change as ‘very much improved’, ‘much improved’, ‘minimally improved’, ‘no change’, ‘minimally worse’, ‘much worse’, or ‘very much worse’.

Participants were asked to return all unused study medication and empty blister packs for drug accountability and controlled destruction. Participants recommenced their regular analgesia or were prescribed codeine if their pain has been effectively controlled at the end of the study period. All study participants were followed up by telephone seven days later to ascertain any late adverse events.

### 2.4. Withdrawal of Participants from Study

Participants could withdraw from study at any time and without reason. It was planned that data would be collected up until the time of withdrawal and used for analysis with the consent of the participant. The investigator could to withdraw participants from the study at any time due to adverse events, violation of the study protocol, lack of efficacy, or loss to follow-up.

### 2.5. Pharmacovigilance and Reporting of Adverse Events

Adverse events were captured verbally through any contact with participants or in pain diaries. Serious adverse events were events resulting in death, a life-threatening situation, hospitalisation, or significant disability or incapacity. Adverse events were defined as any untoward medical occurrence observed in a participant during or following administration of codeine that did not necessarily have a causal relationship with treatment. Adverse events could include unfavourable and unintended signs and symptoms including abnormal laboratory findings that may or may not have a causal relationship with the use of codeine or paracetamol. Adverse events were recorded and assessed, and the clinical course of the adverse event was followed until resolution, stabilisation, or until it was determined that the study treatment or participation was not the cause. Adverse events were categorised according whether they were (‘expected’) or were not (‘unexpected’) listed on the codeine phosphate 30 mg “Summary of Product Characteristics” [45].

### 2.6. Outcome Measures

The primary endpoint in this study was codeine non-response defined as a participant who had <30% reduction in BPI-sf pain ‘on the average’ at the end of the course of codeine therapy. This was measured as the difference between pain ‘on the average’ at pre-codeine baseline (study visit two) and the mean of pain ‘on the average’ during codeine therapy (day 0 to day 4). Secondary endpoints used in this study were *CYP2D6* genotype, measurements of urinary codeine O-demethylation metabolites, patient reported outcomes (BPI-sf, Patient Global Impression of Change to Health) assessor determined outcomes (Clinician Global Impression of Change to Health) and frequency and severity of adverse events.

### 2.7. Data Management and Analysis

The protocol of statistical analyses and calculation of sample size a priori were developed with the biomedical statistical department from Napp Pharmaceuticals. The statistical protocol was peer reviewed by the biomedical statistics department from the University of Leeds as part of the Research Ethics Committee and Medicines and Healthcare Regulatory Agency approval process. The development of the database and statistical analysis of recorded data reported was undertaken by the principal investigator (H.R.). All statistical analyses were performed using IBM SPSS Statistics 21.

#### 2.7.1. Sample Size Calculation

It was calculated that a sample size of 121 participants would give 90% power to detect a larger proportion of codeine non-responders than the null hypothesis of 10%, assuming the true proportion is 20%, using a 5% significance level for a one-sided test. Therefore, a recruitment target of 150 was set based on a drop-out rate of 20% to obtain 121 evaluable participants.

#### 2.7.2. Statistical Analysis

The population for analysis was defined as (I) participants who attended the screening visit (enrolled population) and (II) participants who had their codeine response determined and their CYP2D6 phenotype inferred from genotype (intention-to-treat population).

#### 2.7.3. Analysis of Primary Outcomes

The proportion of codeine non-responders (primary outcome) was calculated with 95% confidence intervals using the exact binomial distribution. A logistic regression analysis was conducted to predict codeine non-response (dependent variable, dichotomous data) from CYP2D6 activity score (independent variable, ordinal data). Analysis of variance (ANOVA) was used to compare log-transformed levels of urinary codeine O-demethylation metabolites between CYP2D6 phenotypes. Logistic regression was used to predict analgesic response using log-transformed urinary total morphine and morphine:creatinine ratio as covariates. A multivariate logistic regression model was developed to predict analgesic response using activity score and log-transformed urinary total morphine and morphine:creatinine ratio. The suitability of the model as predictor of analgesic response was assessed using receiver operating characteristic (ROC) curves.

#### 2.7.4. Analysis of Secondary Outcomes

A descriptive analysis of data from BPI-sf, S-LANSS and Global Impression of Change was performed against codeine non-response and CYP2D6 phenotype. In this study, we used ‘pain on the average’, ‘pain severity’, and ‘pain interference’ in our statistical analysis of BPI-sf outcomes. Concurrent medication was analysed to determine if CYP2D6 substrates impacted on codeine response through autophenocopying. The type and number of substrates was tabulated against CYP2D6 phenotype and codeine response status and a logistic regression analysis used to determine whether CYP2D6 substrates predicted analgesic response. CYP2D6 substrate classification was determined through bioinformatics and cheminformatics databases for each drug identified [46,47]. Study medication (codeine 30 mg QDS and paracetamol 1 g PRN) was excluded from the analysis.

## 3. Results

### 3.1. Characteristics of the Sample Population at Enrolment

A total of 131 participants (age range = 23–79 years, 79 females: 52 males) were enrolled into the study (enrolled population, Figure 1). One male participant withdrew consent following visit one without giving a reason. Three females withdrew from the study due to adverse events. One female (NM phenotype, CYP2D6 activity score 2) withdrew due to moderate rash and difficulty in breathing but also experienced multiple mild adverse events such as nausea, loss of appetite, diarrhoea, restlessness, and disturbed sleep. The other females withdrew due to stomach cramps and disturbed sleep (NM phenotype, CYP2D6 activity score 2) and excessive belching and increased acid reflux (PM phenotype, CYP2D6 activity score 0). The *CYP2D6* genotype of two females could not be determined and they were unable to provide an additional DNA sample for analysis. Thus, 125 participants (age range = 23–78 years, 74 females: 51 males) had their codeine response determined and their CYP2D6 phenotype inferred from genotype (Table 1).

### 3.2. Characteristics of Pain at Enrolment (n = 131)

Data collected for withdrawn participants were included in the analysis of the enrolled group but not the analysis of the ITT group. The mean ± SD duration of pain was 127.51 ± 133.50 months (Median: 73 months, range 9–636 months), with majority of participants diagnosed with low back pain. At enrolment 55/131 (42%) participants had a S-LANSS score ≥12 suggesting a predominantly neuropathic pain state; 76/131 (58%) participants had a S-LANSS score <12 suggesting a predominantly nociceptive pain state.

At enrolment, 125/131 (95%) participants were taking at least one analgesic and/or pain adjuvant (either prescribed or over-the-counter). There were 36/131 (27%) participants taking one analgesic or pain adjuvant, 33/131 (25%) taking two, 30/131 (23%) taking three, 21/131 (16%) taking four, 2/131 (2%) taking five, and 3/131 (2%) taking six. There were 77/131 (59%) participants taking an analgesic prodrug that was reliant on CYP2D6 activity to obtain analgesic efficacy, with co-codamol (codeine and paracetamol combined, 31/131 (24%) participants) and tramadol (26/131 (20%)) being the most commonly prescribed. Only two (1.5%) participants were prescribed more than one CYP2D6 prodrug analgesics concurrently (codeine with tramadol, and co-codamol with tramadol). There were 91/131 (70%) participants that reported they had previously failed to respond to one or more CYP2D6 analgesic prodrugs (50/131 (38%) and 46/131 (35%) respectively), with failure to respond to tramadol being the most commonly reported. There were 14/131 (11%) participants that reported failure to respond to both tramadol and co-codamol and 4/131 (3%) reporting failure to respond to tramadol, codeine, and co-codamol.

### 3.3. CYP2D6 Genotyping and Inferred Phenotype at Enrolment

Two of the 131 saliva samples collected for *CYP2D6* genotyping were discarded because it was not possible to confirm the CYP2D6 activity score. In one sample, it was not possible to classify the CYP2D6 phenotype due to undetermined allele duplication indicating UM if present or NM if absent. In the other sample, it was not possible to determine the presence of the reduced functional allele *41. Phenotype was inferred from 129 of the 131 enrolled participants as follows: PM = 12/129 (9%), IM = 6/129 (5%), NM = 109/129 (85%), and UM = 2/129 (2%). The proportion of these phenotypes was within the range estimated in a general Caucasian population by Crews et al. [21]. The majority of the NM phenotype group (59/129, 46%) possessed full CYP2D6 activity (activity score of 2.0), with 32/129 (25%) having an activity score of 1.0, and 18/129 (14%) having an activity score of 1.5. The most common diplotype was CYP2D6*1/*2 (NM activity score of 2), followed by the fully functional CYP2D6*1/*1 (NM activity score of 2), CYP2D6*1/*4 (NM activity score of 1) and CYP2D6*4xN/*4 (PM activity score of 0).

A risk of suboptimal response to codeine is likely for PM, IM, or UM phenotypes. The number of participants from the enrolled sample that were taking prodrugs reliant on CYP2D6 activity were 10/12 (83%) PMs, 4/6 (67%) IMs, and 1/2 (50%) UMs. The number of participants from the enrolled sample that reported they had failed to respond to prodrug analgesic medication in the past were 9/10 (90%) PMs, 3/6 (50%) IMs, and 1/2 (50%) UMs.

### 3.4. Analysis of CYP2D6 Phenotype and Analgesic Response to Codeine

There were statistically significant reductions in ‘pain severity’, ‘pain interference’, and ‘pain on the average’, between pre-codeine baseline and the end of codeine treatment when measured as the difference between pain ‘on the average’ at pre-codeine baseline (study visit two) and the mean of pain ‘on the average’ during codeine therapy (i.e., day 0 to day 4, Table 2). However, responder analysis revealed that 99/125 (79%) participants that did not achieve a clinically meaningful reduction of pain of ≥30% from baseline and these participants were categorised as codeine non-responders (41/51 (80%) males, 58/74 (78%) females).

There were 53/125 (42%) participants with a S-LANSS score ≥12 suggesting a predominantly neuropathic pain state; 72/125 (58%) participants had a S-LANSS score <12 suggesting a predominantly nociceptive pain state. There were 26/124 (21.0%) participants categorised as codeine responders (i.e., ≥30% reduction of pain) of which 10/26 (39%) had an S-LANSS score <12. Thus, 43/98 (44%) of participants with an S-LANSS score <12 did not respond to codeine. There were no significant differences in the frequency of response between participants categorised as having predominantly neuropathic or nociceptive pain states (*z* = 0.496, *p* = 0.62, odds ratio = 1.25 (95% CI: 0.52, 3.03)).

All PMs, IMs, and UMs were categorised as codeine non-responders (19 of 125 (15%) participants) and were negatively binomial distributed suggesting that these phenotypes would not respond to codeine in the general population (Table 3). There were 106/125 (83%) participants categorised as NMs. There were 80/106 NMs (76%) categorised as codeine non-responders. Of these, 39/56 (70%) NMs with activity scores of 2 (two fully functional alleles) were categorised as codeine non-responders, 14/18 (78%) with activity scores of 1.5 were categorised as codeine non-responders and 27/32 (84%) with activity scores of 1 were categorised as codeine non-responders.

The logistic regression analysis found that CYP2D6 activity score made a significant contribution to the prediction of codeine response (*Beta* = 0.963, Wald = 5.67 *p* = 0.017). The model Chi-square indicated that CYP2D6 activity score affected codeine response (Chi-sq = 6.78, df = 1, *p* = 0.009). The odds ratio (95% confidence interval) was 2.62 (1.186–5.790) suggesting that individuals with a high CYP2D6 activity score were 2.62 times more likely to respond to codeine than individuals with a low CYP2D6 activity score.

### 3.5. CYP2D6 Phenotype and Global Impression of Change

Analysis of the Patient Global Impression of Change to Health found that 20/26 (77%) participants who obtained an analgesic response to codeine reported improvement on the global impression of change questionnaire and 70/99 (71%) participants who did not obtain an analgesic response to codeine reported improvement on the global impression of change questionnaire. All PMs, IMs, and UMs failed to obtain an analgesic response yet 9/11 (82%) PMs reported improvement, 2/6 (33%) IMs reported improvement and 1/2 (50%) UMs reported improvement. Of the NMs who did not obtain an analgesic response 58/80 (73%) reported improvement compared with 20/26 (77%) of participants who did obtain an analgesic response. The Clinician Global Impression of Change to Health was compared with the self-reported Patient Global Impression of Change to Health for accuracy and was successfully matched in 72% of the participants.

### 3.6. Analysis of Adverse Events

There were no reported serious adverse events during the course of the study. The most common ‘expected’ adverse events were headache (46/125 (37%) participants), nausea (41/125 (33%) participants), and constipation (37/125 (30%) participants), with the majority of ‘expected’ adverse events reported by codeine non-responders. The most common ‘unexpected’ adverse events were diarrhoea (13/125 (10%) participants), stomach cramps (11/125 (9%) participants), and flu-like symptoms (11/125 participants (8%)), with a majority of ‘unexpected’ adverse events reported by codeine non-responders. Separate logistic regression analyses of each of these adverse events (i.e. headache, constipation, dry mouth, nausea, and drowsiness) with CYP2D6 activity score and responder status were not statistically significant.

### 3.7. Analysis of CYP2D6 Autophenocopying

Concurrent medication was analysed to determine if CYP2D6 substrates impacted on codeine response through autophenocopying. There were 41/125 (33%) participants not taking CYP2D6 substrates, 47 (38%) taking one CYP2D6 substrate, 23 (18%) taking two CYP2D6 substrates, 10 (8%) taking three CYP2D6 substrates, and 4 (3%) taking four CYP2D6 substrates. Logistic regression analysis found that the number of concurrent CYP2D6 substrates and log urinary transformed total morphine did not make a significant contribution to the prediction of codeine response (number CYP2D6 substrates *Beta* = 0.08, Wald = 0.16, *p* = 0.69; urinary transformed total morphine *Beta* = 0.000155, Wald = 2.829208, *p* = 0.09).

### 3.8. Analysis of Urinary Codeine O-demethylation Metabolites

Urine samples were collected before the first dose of codeine on study visit two (pre-codeine baseline day 0) and after the first dose of codeine on day 4. The sample of one participant was discarded due to leakage in transit from the clinic to the laboratory. Thus, 124 samples were analysed for codeine O-demethylation metabolites (Table 3). Day 4 data for each participant categorised as responder or non-responder for each CYP2D6 phenotype/activity are shown in Figure 2, Figure 3, Figure 4 and Figure 5.

Data for urinary total morphine were log transformed using a base −10 logarithm to reduce skew. One urine sample from day 4 of a participant categorised as a PM was negative for urinary total morphine metabolites (<50 µg/L) and was not included in the log transformed calculation. Log transformed data was categorised according to CYP2D6 phenotype and analgesic response. There were no significant effects of responder status in the different categories of activity score for NM phenotypes (activity score = 1, F(1,29) = 0.12, *p* = 0.73: activity score = 1.5, F(1,16) = 0.59, *p* = 0.45: activity score = 2, F(1,54) = 6.64E-10, *p* = 1.0). There were similar findings for morphine:creatinine ratio calculated to correct for levels of hydration. One-way ANOVA on log transformed morphine:creatinine ratio found significant effects between CYP2D6 activity scores (F(5,117) = 38.43, *p* < 0.001). There were no significant effects of responder status in the different categories of activity score for NM phenotypes (activity score = 1, F(1,28) = 0.03, *p* = 0.87: activity score = 1.5, F(1,16) = 0.24, *p* = 0.63: activity score = 2, F(1,54) = 0.43, *p* = 0.51). Logistic regression analysis found that log transformed total morphine and morphine:creatinine ratio did not make a significant contribution to the prediction of codeine response (total morphine *Beta* = −0.54, Wald = 0.415, *p* = 0.52; morphine:creatinine ratio *Beta* = 1.50, Wald = 2.60, *p* = 0.11).

### 3.9. Predicting Analgesic Response from Codeine Urinary Metabolites

We devised a simple scoring system in an attempt to predict analgesic response to codeine (≥30% relief) for urinary total morphine ranges and morphine:creatinine ratio for different CYP2D6 activity scores based on detected urinary metabolite (Table 4) whereby;
0 points = Unlikely to respond to codeine (activity score = 0 (PM) and activity score = 0.5 (IM), expected urinary total morphine <500 µg/L, expected urinary morphine:creatinine ratio = 0–100 µg/mmol);1 point = Uncertain whether will respond to codeine (activity score = 1.0 (NM), expected urinary total morphine = 500–1499 µg/L, expected urinary morphine:creatinine ratio = 101–250 µg/mmol)2 points = Likely to respond to codeine (activity score = 1.5 or 2.0 (NM), expected urinary total morphine = 1500–7500 µg/L, expected urinary morphine:creatinine ratio = 251–1000 µg/mmol)3 points = Uncertain whether will respond to codeine and potential for adverse events (activity score >2.0 (UM), expected urinary total morphine >7500 µg/L, expected urinary morphine:creatinine ratio >1000 µg/mmol).

A multivariate logistic regression analysis was used to determine whether log-transformed mean urinary total morphine for day 4 could predict analgesic response and a ROC curve used to assess the suitability of the model. The test of the full model against a constant only model was significant at *p* < 0.05 for predicting codeine responder status using urinary total morphine for day 4 (*Beta* = 2.64, Wald = 35.46, *p* < 0.001). The model chi-square goodness of fit was significant (chi-sq. = 43.47, df = 1, *p* < 0.001). Overall prediction success was 79% with an 82% prediction success for expected codeine non-response and 75% prediction success for expected codeine response, with a sensitivity 0.8, specificity 0.78. Mean urinary morphine:creatinine ratio for day 4 the test of the full model against a constant only model was significant at *p* < 0.05 for predicting codeine responder status (*Beta* = 2.77, Wald = 34.42, *p* < 0.001). The model chi-square goodness of fit was significant (chi-sq. = 44.46, df = 1, *p* < 0.001). Overall prediction success was 79% with an 88% prediction success for expected codeine non-response and 68% prediction success for expected codeine response, with a sensitivity of 0.76, and specificity of 0.83.

### 3.10. Predicting CYP2D6 Activity Score from Codeine Urinary Metabolites

We devised a simple scoring system in an attempt to predict CYP2D6 activity score from urinary total morphine and morphine:creatinine ratio based on detected urinary metabolite (Table 5) whereby;
0 points = activity score = 0 (PM), expected urinary total morphine = 0–150 µg/L, expected urinary morphine:creatinine ratio ≤20 µg/mmol);0.5 points = activity score = 0.5 (IM), expected urinary total morphine 151–500 µg/L, expected urinary morphine:creatinine ratio = 21–100 µg/mmol);1 point = activity score = 1.0 (NM), expected urinary total morphine = 501–2000 µg/L, expected urinary morphine:creatinine ratio = 101–300 µg/mmol)1.5 points = activity score = 1.5 (NM), expected urinary total morphine = 2001–3000 µg/L, expected urinary morphine:creatinine ratio = 301–375 µg/mmol)2.0 points = activity score = 2.0 (NM), expected urinary total morphine = 3001–7500 µg/L, expected urinary morphine:creatinine ratio = 376–600 µg/mmol)3.0 points = activity score >2.0 (UM), expected urinary total morphine >7501 µg/L, expected urinary morphine:creatinine ratio >601 µg/mmol).

A multivariate logistic regression analysis was used to determine whether log-transformed urinary total morphine could predict CYP2D6 activity score and a ROC curve used to assess the suitability of the model. Urinary total morphine and urinary Morphine:Creatinine ratio at day 4 did not make a significant contribution to the prediction of CYP2D6 activity score.

## 4. Discussion

This study found that 79% of participants did not achieve a clinically significant reduction of pain of ≥30% four days after commencement of codeine 30 mg QDS. The proportion of phenotypes in the sample were within the range estimated for adult Caucasians [21], with 46% of NMs having a CYP2D6 activity score of 2.0 reflecting full CYP2D6 activity. None of the PMs, IMs, and UMs, and only 24.5% of NMs responded to codeine. CYP2D6 activity score made a significant contribution to the prediction of codeine response with high CYP2D6 activity scores being 2.62 times more likely to respond than low CYP2D6 activity scores. The number of concurrent CYP2D6 substrates and urinary morphine did not make a significant contribution to the prediction of codeine response. We devised a simple scoring system to predict analgesic response (≥30% relief) from day four urinary metabolites and found that this had overall prediction success of 79% (sensitivity 0.8, specificity 0.78) for morphine and 79% (sensitivity 0.76, specificity 0.83) for morphine:creatinine ratio.

At enrolment, 59% of our sample had been prescribed analgesics dependent on CYP2D6 activity such as co-codamol and tramadol, a similar percentage to a recent service improvement project that reviewed prescribing information provided by general practitioners (GP) for new referrals to a UK National Health Service hospital pain clinic [37]. Prescribers are advised to review medication within 2–4 weeks after titration to identify patients that are unlikely to respond [51], although in our sample 70% of participants reported that they had previously failed to respond to one or more CYP2D6 analgesics despite still consuming such medication. Our study found no differences in the number of adverse events (mostly headache, dry mouth, nausea, drowsiness, and constipation) between phenotype and/or activity score. Literature on the incidence of adverse events associated with different phenotypes is sparse and contradictory [52,53,54,55].

In our responder analysis, 79% participants did not achieve a clinically meaningful reduction of pain of ≥30% to codeine 30 mg QDS, based on the cut-point for response consistent with current guidelines from Initiative on Methods, Measurement, and Pain Assessment in Clinical Trials (IMMPACT) [56,57]. Analysis of pre-post change in mean ‘pain severity’, ‘pain interference’, and ‘pain on the average’ of the BPI-sf found statistically significant improvements, although the use of mean data can be misleading because between-group averages may hide proportions of participants that may have responded very well, or extremely poorly to the drug [58]. This is supported by evidence from studies of pharmacological interventions that have found that pain outcomes have U-shaped distributions with some participants experiencing substantial reductions in pain and some minimal improvement [59,60,61,62]. We found that 43% (42/97) of non-responders scored ≥12 on S-LANSS in line with evidence suggesting that neuropathic pain is less likely to respond to codeine than predominantly nociceptive or mixed pain [63]. None of the PMs, IMs, or UMs responded to codeine 30mg QDS [15,17,23], although we recognise that groups samples were small with only two UMs. If the parent drug (codeine and tramadol) must be biotransformed to an active metabolite for therapeutic effect, then UM phenotypes are at risk of experiencing adverse drug reactions from high levels of the active metabolite, with IM and PM phenotypes experiencing no therapeutic effects [15,17,23]. Evidence suggests that UM phenotypes suffer lack of efficacy due to rapid metabolism whereas PM phenotypes suffer from complications from higher than desired plasma concentrations of the drug [64,65]. Ultra-rapid formation of morphine would only short-term analgesia that is not maintained for 4–6 h and would pose a risk of toxicity in UMs [23]. Therefore, codeine is not an appropriate drug for long term analgesia for persistent pain.

It is also possible that a disproportionately high number of treatment resistant participants in the sample was a consequence of polypharmacy. Participants were allowed to maintain concurrent prescribed medication during the study period providing medication was not a CYP2D6 inhibitor (excluding oral contraceptives) to prevent phenocopying through enzyme inhibition. However, five participants were taking CYP2D6 inhibitors with up to a 50% reduction in enzyme activity and all were NMs and codeine non-responders. Three of these participants (activity score = 2) were prescribed oral contraceptives, one (activity score = 2) was prescribed diltiazem (weak inhibitor, [27]), and one (activity score = 1) was self-administrating over-the-counter diphenhydramine (Benadryl^®^, moderate inhibitor [29]). It is likely that CYP2D6 phenocopying occurred in these participants resulting in NMs with CYP2D6 activity scores of 2 presenting as activity scores of 1, and those with activity scores of 1 presenting with activity scores of 0.5 (i.e., as IMs). In addition, three NMs were taking strong CYP2D6 inhibitors (paroxetine or fluoxetine) of which two (activity score = 1.5 and activity score = 2) were codeine non-responders. These individuals down-titrated these CYP2D6 inhibitors so that they had ceased taking the drug three days before the baseline visit, although it is possible that this wash out period was insufficient for systemic clearance of these drugs and their metabolites.

There is debate about the combination of CYP2D6 alleles and level of CYP2D6 activity that constitutes the IM phenotype. We used an activity score of 0.5 to infer IM phenotype in-line with Clinical Pharmacogenetics Implementation Consortium guidelines (1 non-functional allele + 1 reduced function allele: activity score = 0.5). The Dutch Pharmacogenetics Working Group recommend that IM phenotype is inferred from an activity score of 0.5 or an activity score of 1 (fully functional allele + non-functional allele or homozygous partially functional alleles) because of reduced enzyme function compared with NM phenotypes with activity scores of 1.5 or 2 [30,31]. Commercially available *CYP2D6* microarrays such as the Amplichip^®^ infer IM phenotype from activity scores of 0.5 or 1 (homozygous partially functional alleles) because of reduced enzyme function compared with NM phenotypes with activity scores of 1, 1.5, or 2 [66,67,68,69]. If we had categorised IM using an activity score of 0.5–1.0 instead of 0.5 then we would have observed codeine response in 13% participants rather than 0%. If NM was categorised using an activity score of 1.5–2.0 we would have observed a codeine response in 28% of participants rather than 24.5% of participants when categorised using activity scores 1.0–2.0. It is reasonable from these findings to suggest that a CYP2D6 activity score of 1.0 should not be categorised as an NM phenotype because codeine response was only achieved in a minority of participants. However, an activity score of 1.0 should not be categorised IM because of increased CYP2D6 enzyme activity and the possibility of responding to codeine compared with individuals with an activity score of 0.5 who will not respond to codeine. Perhaps a new phenotype classification of moderate metaboliser (MM) may be a more accurate description for an activity score of 1.0. The MM phenotype classification would represent increased enzyme activity compared with the IM phenotype and reduced enzyme activity compared with the NM phenotype.

One should be cautious when extrapolating our findings to the general pain population. The main limitations of the study were the low sample of PM, IM, and UM participants and the poor analgesic response rates in all groups. This impacts on the reliability of the predictive model. It was not possible to enrich the cohort through additional recruitment in the present study, so we recommend that the validity of our scoring system and predictive model is evaluated in a multi-centre study with larger samples. The poor response rate is likely to be due to the fact that participants had been referred to a specialist pain clinic because they were treatment resistant. Eighty individuals were excluded from participation because they had previously experienced intolerable side effects to morphine, codeine, co-codamol, or tramadol. Also, individuals may have declined to participate in the study because of intolerable side effects to codeine in the past. We suspect that these exclusions would be UMs, IMs, and PMs biasing our sample toward individuals with NM and MM phenotypes.

It was assumed that the duration of the wash-out period was sufficient to reduce potential drug interactions with CYP2D6 inhibitors. Five participants continued to take CYP2D6 inhibitors during codeine treatment and this may have reduced their responsiveness to codeine and affected urinary measurements of codeine metabolites. Medication compliance was monitored by diary entries and by counting returned medication/blister packaging at clinic visits and found to be 75–100%, with 82% of the participants categorised as fully compliant. Nevertheless, it is possible that some participants supplemented study medication with over-the-counter codeine at home that is likely to elevate concentrations of codeine metabolites. Likewise, failure to collect urine samples at the time specified by the investigators (i.e., one to two hours after the morning codeine dose) would result in lower than expected concentrations of codeine metabolites. A written reminder to the participant to collect the day 4 urine sample was printed on the day 3 pain diary.

Another confounder was the selection of *CYP2D6* alleles for the customised genotyping microarray as participants may have *CYP2D6* polymorphisms not included in the microarray or that are yet to be discovered. In these individuals, the allele *1 (no polymorphisms detected) would have been allocated indicating a fully functional allele. As scientific advancements in genotyping methodology and knowledge about *CYP2D6* polymorphisms are ongoing, it is recommended that investigators consult the Human *CYP2D6* Allele Nomenclature for an up to date overview of *CYP2D6* alleles when customising *CYP2D6* microarrays [18].

## 5. Conclusions

This study provides tentative evidence that day 4 urinary codeine O-demethylation metabolites could predict non-response following a short course of codeine and could be utilised in the clinical assessment of codeine response at the point of care to improve analgesic efficacy and safety in codeine therapy. We offer a scoring system to predict codeine response from urinary morphine and urinary morphine:creatinine ratio collected on the morning of day 4 of codeine 30 mg QDS but this requires validation before it could be considered for use to assess codeine response in clinical practice. Translating *CYP2D6* genotyping and/or phenotyping to clinical reality remains problematic. Whether a urine test on day 4 has clinical utility is questionable as the response at this time-point is detectable by patient self-report. Clearly, a pre-emptive urine test is desirable. Moreover, urinary concentrations of morphine may be more useful than genotype when inferring phenotype in clinical practice. There remains a need for higher standard pharmacogenomic education especially on CYP2D6 and pain management in undergraduate and post-graduate education that is relevant to clinical practice for all disciplines.

An adequately sized multi-centre study conducted in a primary care setting using samples from the general population that includes *CYP2D6* genotyping is needed to confirm our findings. Future pharmacogenomic studies using codeine in persistent pain patient populations could offer genotyping to participants who have declined to take part because of codeine intolerability. This would enable a full analysis of the frequencies of CYP2D6 phenotypes. There is also a need for a multi-centre study to validate the concentration range of codeine O-demethylation metabolites against activity scores. We recommended that participants with a non-functional allele + fully functional allele (activity score = 1) are analysed separately to those with homozygous partially functional alleles (activity score = 1) to determine whether there is a difference between the metabolic capabilities of these individuals. This would aid development of an accurate method to infer CYP2D6 phenotype in a clinical setting.

## Figures and Tables

**Figure 1 medicina-55-00220-f001:**
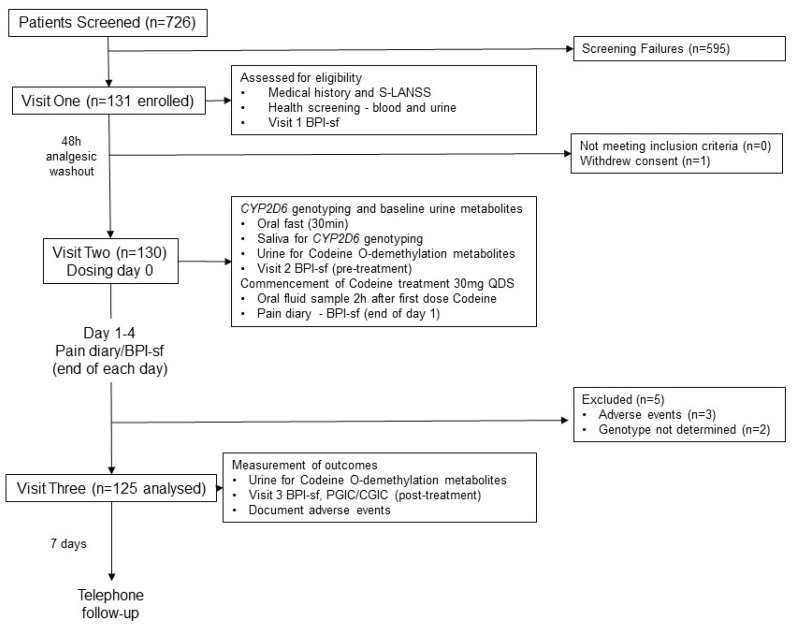
Flow diagram of the progress of participants through phases of the study, adapted from the Consolidated Standards of Reporting Trials (CONSORT) template [48]. PGIC, Patient Global Impression of Change to Health; CGIC, Clinician Global Impression of Change to Health.

**Figure 2 medicina-55-00220-f002:**
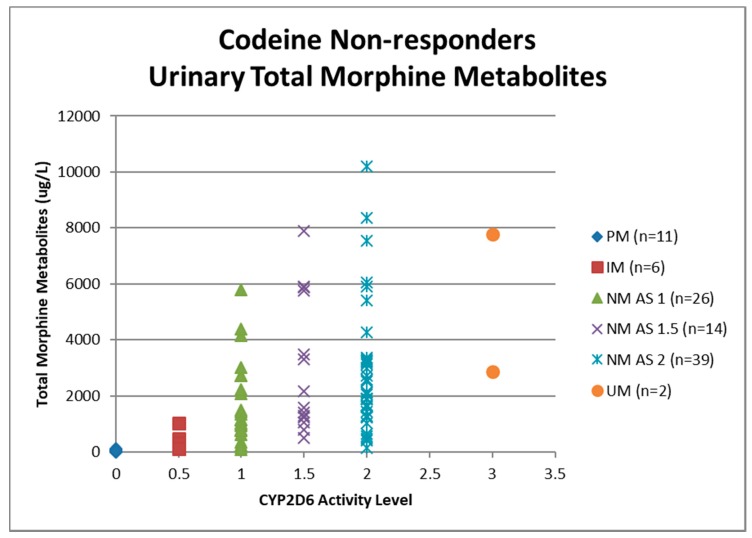
Individual value plot of day 4 urinary total morphine metabolites for each CYP2D6 phenotype/activity score of participants defined as codeine non-responders (<30% reduction in mean day 0–day 4 ‘average pain’ 0–10 numerical rating scale (NRS) score when compared to baseline). AS, Activity Score; PM, Poor metaboliser; IM, Intermediate metaboliser; NM, Normal metaboliser; UM, Ultra metaboliser.

**Figure 3 medicina-55-00220-f003:**
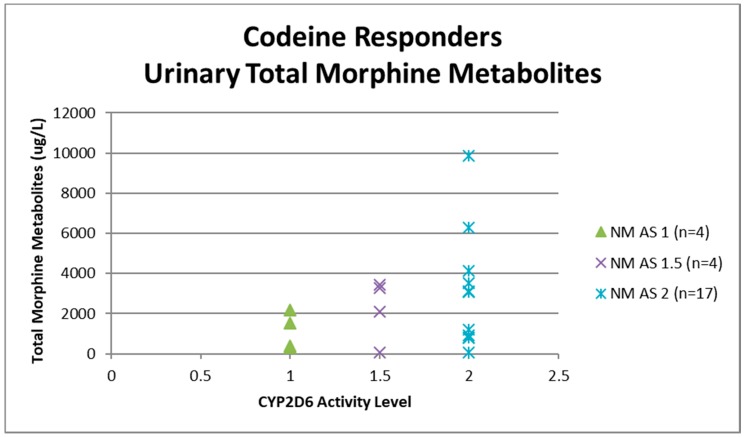
Individual value plot of day 4 urinary total morphine metabolites for each CYP2D6 phenotype/activity score of participants defined as codeine responders (≥30% reduction in mean day 0–day 4 ‘average pain’ 0–10 numerical rating scale (NRS) score when compared to baseline). AS, Activity Score; NM, Normal metaboliser.

**Figure 4 medicina-55-00220-f004:**
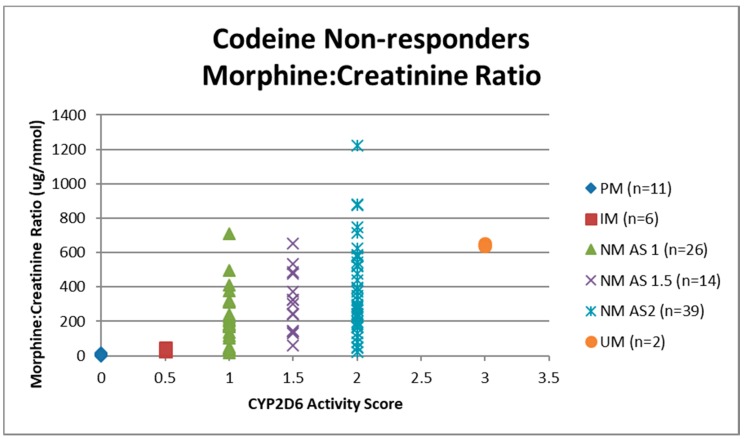
Individual value plot of day 4 urinary morphine:creatinine ratio for each CYP2D6 phenotype/activity score of participants defined as codeine non-responders (<30% reduction in mean day 0–day 4 ‘average pain’ 0–10 numerical rating scale (NRS) score when compared to baseline). AS, Activity Score; PM, Poor metaboliser; IM, Intermediate metaboliser; NM, Normal metaboliser; UM, Ultra metaboliser.

**Figure 5 medicina-55-00220-f005:**
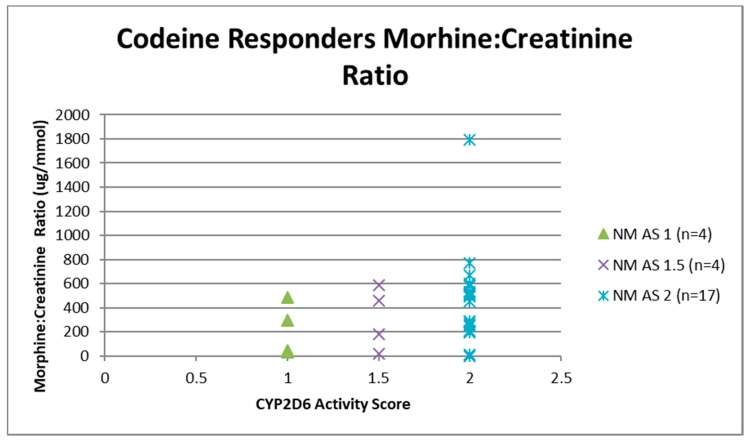
Individual value plot of day 4 urinary morphine:creatinine ratio for each CYP2D6 phenotype/activity score of participants defined as codeine responders (≥30% reduction in mean day 0–day 4 ‘average pain’ 0–10 numerical rating scale (NRS) score when compared to baseline). AS, Activity Score; NM, Normal metaboliser.

**Table 1 medicina-55-00220-t001:** Characteristics of the study from the intention to treat analysis (*n* = 125) participants at pre-codeine baseline day 0.

Variable	ITT Group		
	Male	Female	Total
**Number (%)**	51 (40.8%)	74 (59.2%)	125 (100%)
**Age (minimum–maximum)**	27–78 years	23–77 years	23–79 years
**Age (Mean ± SD)**	57.10 ± 12.33	55.64 ± 14.74	56.26 ± 13.71
**Mean ± SD BPI-sf**			
Pain severity	5.73 ± 1.61	6.25 ± 1.82	6.09 ± 1.75
Pain interference	4.99 ± 2.68	5.05 ± 2.59	5.02 ± 2.51
Pain ‘on the average’	5.46 ± 1.95	5.81 ± 2.21	6.31 ± 1.76
**Tally (%) S-LANSS**			
≥12—Neuropathic pain	18/51 (35.30%)	35/74 (47.30%)	53/125 (42.74%)
<12—Nociceptive pain	33/51 (64.70%)	39/74 (52.71%)	71/125 (57.26%)
**Medication at enrolment**			
Prescribed CYP2D6 analgesic prodrug	29/51(56.86%)	44/74(59.46%)	73/125(58.4%)
**Tally (%) Phenotype**			
PM (Activity Score = 0.0)	4/51 (7.84%)	7/74 (9.49%)	11/125 (8.8%)
IM (Activity Score = 0.5)	5/51 (9.80%)	1/74 (1.35%)	6/125 (4.8%)
NM (Activity Score = 1.0–2.0)			
NM (Activity Score = 1.0)	12/51 (23.53%)	20/74 (27.03%)	32/125 (25.6%)
NM (Activity Score = 1.5)	9/51 (17.65%)	9/74 (12.16%)	18/125 (14.4%)
NM (Activity Score = 2.0)	20/51 (39.22%)	36/74 (48.65%)	56/125 (44.8%)
UM (>2.0)	1/51 (1.96%)	1/74 (1.35%)	2/125 (1.6%)

ITT, Intention to treat; SD, Standard Deviation; BPI-sf, Brief Pain Inventory short form; NRS, Numerical Rating Scale; S-LANSS, Self-report version of the Leeds Assessment of Neuropathic Pain Scale; PM, Poor metaboliser; IM, Intermediate metaboliser; NM, Normal metaboliser; UM, Ultra metaboliser.

**Table 2 medicina-55-00220-t002:** Mean ± SD BPI-sf scores in the ITT population (*n* = 125).

BPI-sf	Visit 1 Pre-Analgesic Washout	Visit 2 Pre-Codeine Baseline Day 0	End of Dosing Day (Day 0)	End of Day 1	End of Day 2	End of Day 3	End of Day 4	Mean Day 0–4
Pain severity (composite)	6.31 ± 1.76	6.09 ± 1.75	5.93 ± 1.99	5.67 ± 2.27	5.54 ± 2.28	5.57 ± 2.40	5.62 ± 2.45	5.67±2.17 *
Pain interference (composite)	5.02 ± 2.51	6.14 ± 2.11	5.31 ± 2.40	5.09 ± 2.58	4.90 ± 2.59	4.91 ± 2.66	4.91 ± 2.73	5.02 ± 2.51 *
Pain ‘on the average’	6.09 ± 1.75	6.31 ± 1.76	6.00 ± 1.94	5.68 ± 2.28	5.55 ± 2.24	5.51 ± 2.37	5.57 ± 2.35	5.66 ± 2.11 *

The pain severity score was calculated as the mean of NRS scores for ‘worst’, ‘least’, ‘average’ and ‘right now’ pain. The pain interference score was calculated as the mean of seven items of pain interference. * *p* < 0.05 indicating statistical significance between mean day 0–4 and pre-codeine. SD, Standard Deviation; BPI-sf, Brief Pain Inventory short form; ITT, Intention to treat; NRS, Numerical Rating Scale.

**Table 3 medicina-55-00220-t003:** Urinary codeine O-demethylation metabolites according to CYP2D6 activity score (AS) and codeine response.

Sample	PM (AS 0.0)	IM (AS 0.5)	NM (AS 1.0–2.0)	NM (AS 1.0)	NM (AS 1.5)	NM (AS 2.0)	UM (>2.0)
Total number (% non-response within phenotype)	11 (100%)	6 (100%)	96 (83.33%)	32 (84.38%)	18 (77.78%)	56 (69.64%)	2 (100%)
Codeine Responders	0	0	26	5	4	17	0
Codeine non-responders	11	6	80	27	14	39	2
Mean ± SD total morphine (µg/L)							
Codeine responders	no responders identified	no responders identified	2902.65 ± 2504.43	1754.6 ± 1661.56	2523.75 ± 1634.72	3329.47 ± 2822.71	no responders identified
Codeine non-responders ^#^	44.18 ± 39.66	415.16 ± 317.83	2419.77 ± 2106.61	1444.65 ± 1342.90	2925.43 ± 2410.52	2846.08 ± 2226.72	5330.00 ± 3464.82
Mean ± SD morphine:creatinine ratio (µg/L)							
Codeine responders	no responders identified	no responders identified	420.04 ± 346.91	271.80 ± 228.06	347.00 ± 243.93	480.82 ± 389.56	no responders identified
Codeine non-responders ^#^	6.91 ± 4.46	35.83 ± 7.14	311.94 ± 231.47	197.96 ± 150.23	307.00 ± 177.35	389.69 ± 263.18	646.50 ± 12.02

^#^ one sample missing from NM (AS 1.0), *n* = 26 not 27. SD, Standard Deviation; PM, Poor metaboliser; IM, Intermediate metaboliser; NM, Normal metaboliser; UM, Ultra metaboliser.

**Table 4 medicina-55-00220-t004:** Points system for predicting codeine response from day 4 urinary total morphine metabolites (µg/L) based on detected urinary metabolite.

CYP2D6 Activity Score (AS)	CYP2D6 Phenotype	Codeine Non-Responder	Codeine Responder	Suggested Range for Predicting Codeine Response	Expected Codeine Response	Points	Rationale
		Mean total morphine (µg/L)	Mean total morphine (µg/L)	Mean total morphine (µg/L)			
0	PM	44.2	no responders identified	0–499	Probably will not respond to codeine	0	Includes PM and IM phenotypes
0.5	IM	415.2	no responders identified	0–499	Probably will not respond to codeine	0	Includes PM and IM phenotypes
1	NM	1444.7	1754.6	500–1499	May not respond to codeine	1	Includes NM AS 1 phenotype or individuals CYP2D6 phenocopying
1.5	NM	2925.4	2523	1500–7500	Should respond to codeine	2	Includes NM AS 1.5 and AS 2 phenotypes: Expected to respond to codeine
2	NM	2846.1	3329.5	1500–7500	Should respond to codeine	2	Includes NM AS 1.5 and AS 2 phenotypes: Expected to respond to codeine
3	UM	5330	no responders identified	>7500	May not respond and potential ADRs	3	Includes UM phenotypes: Potential for ADRS
		Mean morphine: creatinine ratio (µg/mmol)	Mean morphine: creatinine ratio (µg/mmol)	Mean morphine: creatinine ratio (µg/mmol)			
0	PM	6.91	no responders identified	0–100	Probably will not respond to codeine	0	Includes PM and IM phenotypes
0.5	IM	35.83	no responders identified	0–100	Probably will not respond to codeine	0	Includes PM and IM phenotypes
1	NM	197.96	271.8	101–250	May not respond to codeine	1	Includes NM AS 1 phenotype or individuals CYP2D6 phenocopying
1.5	NM	307	347	251–1000	Should respond to codeine	2	Includes NM AS 1.5 and AS 2 phenotypes: Expected to respond to codeine
2	NM	389.69	480.82	251–1000	Should respond to codeine	2	Includes NM AS 1.5 and AS 2 phenotypes: Expected to respond to codeine
3	UM	646.5	no responders identified	>1000	May not respond and potential ADRs	3	Includes UM phenotypes: Potential for ADRs

PM, Poor metaboliser; IM, Intermediate metaboliser; NM, Normal metaboliser; UM, Ultra metaboliser; AS, Activity score; ADR, Adverse drug reaction.

**Table 5 medicina-55-00220-t005:** Points system for predicting for predicting CYP2D6 activity score from day 4 urinary total morphine metabolites (µg/L) and morphine:creatinine ratio (µg/mmol) based on detected urinary metabolite.

CYP2D6 Activity Score (AS)	CYP2D6 Phenotype	Codeine Non-Responder	Codeine Responder	Suggested Range for Predicting CYP2D6 Activity Score	Points	Rationale
		Mean total morphine (µg/L)	Mean total morphine (µg/L)	Mean total morphine (µg/L)		
0	PM	44.2	no responders identified	0–150	0	<300 µg/L is a negative cut off point in codeine drug screens [49]
0.5	IM	415.2	no responders identified	151–500	0.5	Severely reduced function and comparable to PMs [50]
1	NM	1444.7	1754.6	501–2000	1	Expected to respond to codeine
1.5	NM	2925.4	2523	2001–3000	1.5	
2	NM	2846.1	3329.5	3001–7500	2	
3	UM	5330	no responders identified	>7501	3	Potential for ADRs
		Mean morphine: creatinine ratio (µg/L)	Mean morphine: creatinine ratio (µg/L)	Morphine: creatinine ratio (µg/L)		
0	PM	6.91	no responders identified	≤20	0	<300 µg/L is classed as a negative cut off point in codeine drug screens [49]
0.5	IM	35.83	no responders identified	21–100	0.5	Severely reduced function and comparable to PMs [50]
1	NM	197.96	271.8	101–300	1	Expected to respond to codeine
1.5	NM	307	347	301–375	1.5	
2	NM	389.69	480.82	376–600	2	
3	UM	646.5	no responders identified	>601	3	Potential for ADRs

PM, Poor metaboliser; IM, Intermediate metaboliser; NM, Normal metaboliser; UM, Ultra metaboliser; AS, Activity score; ADR, Adverse drug reaction.

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
