# Peer review of "A Single Site Population Study to Investigate CYP2D6 Phenotype of Patients with Persistent Non-Malignant Pain"

_medicina, 2019, doi:10.3390/medicina55060220_

Round 1
Reviewer 1 Report
This manuscript by Radford and colleagues attempt to develop a scoring system to predict Codeine non-response in the patients suffering pain. They screened the 131 patients recruited into this study and suggested the patients to stop their current pain drugs. Then they harvested the saliva from patients for CYP2D6 genotyping, analyzed the urinary Codeine O-demethylation metabolites, and tested the oral fluid O-demethylation Codeine metabolites after treatment of Codeine.
Overall, this is well-designed study, the methods are described clearly, and the data are presented appropriately. Only a few minor revisions need to be done for publish.
Minor Essential Revisions
1) In line 184-185, how long were the saliva samples stored at RT?
2) Provide product information the equipment and solutions used in this study, e.g., LC-MS/MS, the standards used for the Codeine O-demethylation metabolites analysis
Author Response
Reviewer 1
This manuscript by Radford and colleagues attempt to develop a scoring system to predict Codeine non-response in the patients suffering pain. They screened the 131 patients recruited into this study and suggested the patients to stop their current pain drugs. Then they harvested the saliva from patients for CYP2D6 genotyping, analyzed the urinary Codeine O-demethylation metabolites, and tested the oral fluid O-demethylation Codeine metabolites after treatment of Codeine. Overall, this is well-designed study, the methods are described clearly, and the data are presented appropriately. Only a few minor revisions need to be done for publish.
Authors’ Response: Thank you for taking the time to critique our study. We agree with all of your comments and have attempted to resolve each comment in turn. We have amended the manuscript as appropriate and believe that this has increased the quality of our report. We hope that this meets with your approval.
Minor Essential Revisions
1) In line 184-185, how long were the saliva samples stored at RT?
Authors’ Response: We have amended the sentence to clarify. Samples were stored for no more than 6 months depending on recruitment rates – we ran analyses in batches of approximately 30 samples. Samples have stability for up to 5 years - https://www.dnagenotek.com/ROW/pdf/PD-WP-005.pdf
2) Provide product information the equipment and solutions used in this study, e.g., LC-MS/MS, the standards used for the Codeine O-demethylation metabolites analysis
Authors’ Response: We have added this information to the Methods section ‘Collection and Analysis of Urinary Codeine O-demethylation Metabolites’
Reviewer 2 Report
The study is based on short treatment of codeine in patients with persistent non-malignant pain and relation of its efficacy to CYP2D6 genotype and urinary biomarkers. The topic is interesting and the manuscript is well written. However, the study suffers from small sample size and at some stages unclear presentation as well as some major limitations (see points 8-9).
Major comments:
1) The title and aims as well as the primary endpoint of the manuscript suggest that the main topic is the genotypes/phenotypes of non-responders to codeine. This, however, does not translate to results (which are focused on codeine response and genotype or urinary morphine). At least the title could be more informative (consider removing part “Who Do Not Respond to Oral Codeine”.
2) Nomenclature issues: rather use recommended “normal metabolizer (NM)” than previously recommended EM. Additionally, wild-type should not be used in terms of human subjects.
3) Materials and Methods: This section is very lengthy and is formatted similarly to reports on investigational medicinal product rather than a scientific report. Even though it is good to document all the details, consider reducing redundant text (e.g. detailed description of study visits, including their length, or definition of ADRs).
4) Results: The genotype data and the urinary metabolite data is somewhat unclearly presented. Firstly, different figures have different colours for same AS. Secondly, it is not very clear in 3.9 (page 14) what is actually included in the scoring system in the model, the genotype activity score or the detected urinary metabolite? Consider also adding a single figure where non-responders and responders are presented with different symbols (i.e. combine figures 2 and 3 and 4 and 5).
5) Tables 2 and 3: the concentrations for codeine responders in PM, IM and UM groups should be “not available” rather than zero, as there is no data available.
6) Table 4: Refer to comment 4, what is used to predict what? Also consider that in the study group, the actual response rate was poor in all groups, reducing the validity of the scoring system altogether.
7) Discussion, page 17, line 575. Please provide further elaboration on the topic of UMs being at risk of non-response (this claim is presented also earlier in the text). Only single ref 23 is provided in this context. It is clear that UMs are at risk of ADRs, but it seems that the authors may have over-interpreted their findings in group of n=2.
8) Discussion. Moreover, the discussion should mention the limitation of sample size especially with regards to PM, IM and UM groups more clearly.
9) The usefulness of the urine test on day 4 is questionable in the sense that it is not pre-emptive test but rather done at the point where the response could be detected by the patient. Furthermore, rather than seeing other drugs interfering with CYP2D6 as a limitation, the urinary concentrations of morphine could be more useful than genotype in inferring the phenotype in these situations.
Minor comments:
- page 14: it is quite clear that the wash out for at least fluoxetine was not long enough (t1/2 of 4-6 days for fluoxetine and even longer for its metabolite)
Author Response
Reviewer 2
Comments and Suggestions for Authors
The study is based on short treatment of codeine in patients with persistent non-malignant pain and relation of its efficacy to CYP2D6 genotype and urinary biomarkers. The topic is interesting and the manuscript is well written. However, the study suffers from small sample size and at some stages unclear presentation as well as some major limitations (see points 8-9).
Authors’ Response: Thank you for taking the time to critique our study and for identifying areas for improving the clarity of reporting. We agree that the sample size appears small, although our study has a relatively large pain population sample compared with other in this field. We accept all of your comments and have attempted to resolve each comment in turn. We have amended the manuscript as appropriate and believe that this has increased the quality of our report. We hope that this meets with your approval.
Major comments:
1) The title and aims as well as the primary endpoint of the manuscript suggest that the main topic is the genotypes/phenotypes of non-responders to codeine. This, however, does not translate to results (which are focused on codeine response and genotype or urinary morphine). At least the title could be more informative (consider removing part “Who Do Not Respond to Oral Codeine”.
Authors’ Response: We have deleted “Who Do Not Respond to Oral Codeine” from the title
2) Nomenclature issues: rather use recommended “normal metabolizer (NM)” than previously recommended EM. Additionally, wild-type should not be used in terms of human subjects.
Authors’ Response: Thank you for pointing this out. We agree that there has been a shift away from the use of the term extensive metabolizer in recent years and have amended the terms EM and wild type throughout the manuscript.
3) Materials and Methods: This section is very lengthy and is formatted similarly to reports on investigational medicinal product rather than a scientific report. Even though it is good to document all the details, consider reducing redundant text (e.g. detailed description of study visits, including their length, or definition of ADRs).
Authors’ Response: We respect this comment and have made some edits. We believe that some of the operational detail is important, especially when the readership may not have specialism in pain and/or relevant trial methodology. When this type of information is absent from study reports (sometimes due to Editorial space limitations) it hinders operational replication and the ability to critique methodology, for example for systematic reviews. We would consider including some information in a supplementary appendix if pressed.
4) Results: The genotype data and the urinary metabolite data is somewhat unclearly presented.
Firstly, different figures have different colors for same AS.
Authors’ Response: Thank you. We have made colors and symbols consistent across the figures and have modified the legend to reflect NM notation rather than EM
Secondly, it is not very clear in 3.9 (page 14) what is actually included in the scoring system in the model, the genotype activity score or the detected urinary metabolite?
Authors’ Response: This was detected urinary metabolite and we have amended the text to clarify for this section and in section 3.10
Consider also adding a single figure where non-responders and responders are presented with different symbols (i.e. combine figures 2 and 3 and 4 and 5).
Authors’ Response: We have previously considered doing this and found it to be cluttered losing clarity in communication. We would prefer to keep the figures separate if at all possible.
5) Tables 2 and 3: the concentrations for codeine responders in PM, IM and UM groups should be “not available” rather than zero, as there is no data available.
Authors’ Response: Thank you for identifying this oversight. We have inserted ‘no responders identified’ at the appropriate points in Tables 3-5
6) Table 4: Refer to comment 4, what is used to predict what?
Authors’ Response: We believe this was clear in the legend “… for predicting codeine response from day 4 urinary total morphine metabolites (ug/L).” but we have added ‘based on detected urinary metabolite’ to further emphasize the point to Legends for Tables 4 and 5
Also consider that in the study group, the actual response rate was poor in all groups, reducing the validity of the scoring system altogether.
Authors’ Response: We agree and have amended the Discussion to identify this as a limitation of the study
7) Discussion, page 17, line 575. Please provide further elaboration on the topic of UMs being at risk of non-response (this claim is presented also earlier in the text). Only single ref 23 is provided in this context. It is clear that UMs are at risk of ADRs, but it seems that the authors may have over-interpreted their findings in group of n=2.
Authors’ Response: We have amended the Discussion to elaborate on UMs being at risk of non-response and have included additional references
8) Discussion. Moreover, the discussion should mention the limitation of sample size especially with regards to PM, IM and UM groups more clearly.
Authors’ Response: We agree and have amended the Discussion to identify this as a limitation of the study
9) The usefulness of the urine test on day 4 is questionable in the sense that it is not pre-emptive test but rather done at the point where the response could be detected by the patient. Furthermore, rather than seeing other drugs interfering with CYP2D6 as a limitation, the urinary concentrations of morphine could be more useful than genotype in inferring the phenotype in these situations.
Authors’ Response: We agree – thank you for pointing out this omission. We have added some statements to the conclusion about clinical utility.
Minor comments:
- page 14: it is quite clear that the wash out for at least fluoxetine was not long enough (t1/2 of 4-6 days for fluoxetine and even longer for its metabolite)
Authors’ Response: We agree and had mentioned this in the Discussion section (approximately line 625).
Round 2
Reviewer 2 Report
All the raised points have been answered adequately.